# Evaluation of SARS-CoV-2 Seroprevalence and Variant Distribution During the Delta–Omicron Transmission Waves in Greater Accra, Ghana, 2021

**DOI:** 10.3390/v17040487

**Published:** 2025-03-28

**Authors:** Elvis Suatey Lomotey, Jewelna Akorli, Millicent Opoku, Daniel Adjei Odumang, Kojo Nketia, Emmanuel Frimpong Gyekye, Kojo Mensah Sedzro, Nana Efua Andoh, Yvonne Ashong, Benjamin Abuaku, Kwadwo A. Koram, Irene Owusu Donkor

**Affiliations:** 1Department of Epidemiology, Noguchi Memorial Institute for Medical Research, University of Ghana, Legon, Accra P.O. Box LG 581, Ghana; elomotey@noguchi.ug.edu.gh (E.S.L.); doadjei@noguchi.ug.edu.gh (D.A.O.); egyekye@noguchi.ug.edu.gh (E.F.G.); kmensahsedzro@noguchi.ug.edu.gh (K.M.S.); babuaku@noguchi.ug.edu.gh (B.A.); kkoram@noguchi.ug.edu.gh (K.A.K.); 2Department of Parasitology, Noguchi Memorial Institute for Medical Research, University of Ghana, Legon, Accra P.O. Box LG 581, Ghana; jakorli@noguchi.ug.edu.gh (J.A.); mopoku@noguchi.ug.edu.gh (M.O.); kojonke@gmail.com (K.N.); nandoh@noguchi.ug.edu.gh (N.E.A.); yashong@noguchi.ug.edu.gh (Y.A.)

**Keywords:** SARS-CoV-2, sensitivity, seroprevalence, Omicron, Delta, infection

## Abstract

A significant proportion of SARS-CoV-2 infections in Africa were identified as asymptomatic. With the surge of the Omicron variant, asymptomatic participants in epidemiological surveys were key to accurately estimating seroprevalence and true infections in the population. This study assessed seroprevalence, active infections, and circulating variants in Accra, Ghana, during the Omicron wave. Secondary objectives included assessing the association between seroprevalence and sociodemographic factors, vaccination, and adherence to recommended SARS-CoV-2 prevention and control measures. We conducted a cross-sectional survey in Greater Accra in December 2021 using a standardized questionnaire. Serum and naso-oropharyngeal swab samples were collected from 1027 individuals aged ≥ 5 years for the estimation of total antibodies and detection of infection. The study found an overall seroprevalence of 86.8% [95% CI: 84.53–88.77]. PCR test positivity of SARS-CoV-2 was 10%, with the Omicron and Delta variants accounting for 44.1% and 8.8% of infections, respectively. Vaccination (cOR = 10.5, 95% CI: 4.97–26.9, *p* < 0.001) and older age, particularly the 60+ age group (cOR = 6.05, 95% CI: 2.44–20.2, *p* < 0.001), were associated with an increase in odds of seropositivity among participants. High seropositivity of SARS-CoV-2 in Accra was an indication of high exposure and transmission rates and/or high vaccine-induced seroprevalence.

## 1. Introduction

The Severe Acute Respiratory Syndrome Coronavirus-2 (SARS-CoV-2) has been associated with 777,385,370 reported cases and 7,088,757 deaths since its emergence in 2019 [1,2]. Although public health and social measures to prevent the spread of the virus have been significantly eased globally, active infections continue to occur [3,4]. The African continent recorded the fewest cases and lowest mortality throughout the COVID-19 pandemic and continues to report relatively low numbers in its ongoing management phase. However, emerging research indicates considerable serological evidence suggesting that SARS-CoV-2 infection rates in Africa were ultimately comparable to those in other continents [5,6]. The World Health Organization (WHO)’s reports and other relevant studies showed that two-thirds of the African population were exposed to the SARS-CoV-2 virus, with more than 60% of patients being asymptomatic. The wide spread of asymptomatic infections, coupled with limited testing and underreporting, likely contributed to the discrepancy between reported cases and true infection rates [7,8,9,10]. Based on the trajectory of recorded deaths and the relative contagiousness of distinct SARS-CoV-2 variants, the progression of the pandemic in Africa showed a markedly different pattern compared to other parts of the world [5]. The successive waves of infection have been attributed to point mutations in the virus, classified as “variants of concern” that have superseded the wild-type strain [11,12]. The Delta and Omicron variants have been the most dominant variants in Africa [13]. Studies have indicated that Omicron is associated with high transmissibility coupled with minimal severity of illness globally [13,14] compared to Delta, which is associated with a higher disease burden [15]. The first confirmed cases of COVID-19 in Ghana were recorded in March 2020, soon after the WHO declared COVID-19 a global pandemic. The disease spread quickly through the population, with the majority of cases reported in the two most populated cities in Ghana, Accra and Kumasi [16]. These areas recorded high numbers of asymptomatic individuals who had no travel history, indicating local transmission, which led to several public health and social measures, such as lockdown and closing of air and land borders to reduce spread from March to August 2020 [7,17]. Ghana experienced four pandemic waves with epidemic peaks. Each wave was associated with a specific variant of concern. Data available show that the B.1.1 lineage drove local transmission in the first wave, while the second wave was driven by the Alpha variant. The B.1.1.318 variant dominated transmission from April to June 2021, and it was replaced by the Delta variants in August 2021 and, finally, the Omicron variants, whose epidemic peak was the highest nationwide in December 2021 [18,19]. As of 7 April 2024, Ghana had detected a total of 172,075 cases with COVID-19 and 1462 deaths [20]. COVID-19 control measures in Ghana evolved with the global pandemic. Ghana’s international airports and all land borders were closed to international travel on 22 March 2020, followed by a partial lockdown of two major cities, Accra and Kumasi, from 30 March to 22 April 2020. This was coupled with enhanced testing and contact tracing to track community spread. The airport was reopened to international travel on 1 September 2020, with twofold containment measures: (1) travelers must show proof of a negative COVID-19 test (taken at most 72 h before arrival) and (2) travelers must be negative for the SARS-CoV-2 antigen test upon arrival at the Kotoka International Airport (KIA) [18]. A seroprevalence study in the last and first quarters of 2020 and 2021, respectively, showed increasing seropositivity (9.3–19.3%) in the two regions where lockdown was imposed [17]. Thereafter, significant COVID-19-related events occurred, including the emergence of highly transmissible variants of the virus, i.e., Delta and Omicron, and vaccination against SARS-CoV-2, which was expected to increase seropositivity. Currently, 28,515,854 vaccine doses have been administered to the Ghanaian population. A total of 11,782,609 people have received full doses, making up 56.9% of the target population of 20.7 million. Furthermore, 10,545.038 people have received at least one dose of ChAdOx1 nCoV-19 (University of Oxford/AstraZeneca), as reported in the last quarter of 2023. Other deployed vaccines in Ghana included Sputnik V (Gam-COVID-Vac) and the messenger RNA (mRNA) vaccines mRNA-1273 (Moderna), BNT162b2 (Pfizer-BioNTech), and Ad26.CoV2.S (Janssen) [18,20].

According to the WHO, infection dynamics and disease burden are significant focus points of operational research to assess key epidemiological features of emerging pathogens [21]. This study aimed to estimate the extent of infection, seroprevalence, and circulating variants of concern in Greater Accra, Ghana during the fourth wave of infections. Secondary objectives included assessing the associations between seroprevalence or PCR positivity and sociodemographic factors, vaccination, and adherence to recommended SARS-CoV-2 prevention and control measures.

## 2. Materials and Methods

### 2.1. Study Design

This was a cross-sectional study conducted in Greater Accra from 5 December to 19 December 2021. Households were selected from the National Sampling Frame from the Ghana Statistical Service. Using the Kish selection grid and the left-hand rule [22], 20 households were selected from each enumeration area (EA). From each household, two people who consented to participate in the study were included. Sociodemographic information, vaccination status, and health status were obtained from all consenting participants using a standardized questionnaire. The study was aligned with the WHO Unity Studies Protocol [21].

### 2.2. Study Setting, Eligibility, and Sampling

This study was conducted in the Greater Accra region of Ghana. The region is made up 26 districts, with Accra as the regional capital. The region has an estimated population of 5,455,692 according to the 2021 Population Census and serves as the gateway into the country via air travel due to the presence of the Kotoka International Airport. Ghana recorded its first case of SARS-CoV-2 on 12 March 2020 among travelers in Accra. Soon after, there was evidence of local spread of the virus in the region. Persons using the airport were subjected to mandatory SARS-CoV-2 testing and quarantine before entry into the country. The region houses various COVID-19 treatment and testing centres, including the Noguchi Memorial Institute for Medical Research (NMIMR), the National Public Health Reference Laboratory, the Korle Bu Teaching Hospital, and the Ghana Infectious Disease Centre. This study focused on Greater Accra because it is the most densely populated region and it was the region with the first affected city and in which the majority of SARS-CoV-2 cases were detected. Persons aged ≥ 5 years in the Greater Accra Region, Ghana were eligible for recruitment into the study. Travelers and individuals with acute illness were excluded from the study. A total of 1027 participants were recruited, consented, and were sampled. For each household assessed, individuals were stratified into five age groups (5–9, 10–19, 20–39, 40–59, and 60+). Two people from each household who consented were selected for inclusion.

### 2.3. Sample Collection

First, 5 mL of whole blood was collected from each participant as per the WHO guidelines on venipuncture and divided into serum separator and EDTA tubes. Naso- and oropharyngeal swabs were collected from each participant according to standard operating procedure [23] and transferred to a viral transport medium (VTM). The EDTA tubes and swabs were transported on ice, while the serum separator tubes (SSTs) were transported at room temperature. All samples were transported in a cold chain to the laboratory at NMIMR for processing and laboratory analysis. All samples were processed the same day and stored at −20 °C.

### 2.4. Detection of (IgG/IgM) Antibodies Against SARS-CoV-2

The WANTAI SARS-CoV-2-Ab kit (ref. WS-1096), Beijing WANTAI Biological Pharmacy Enterprise Co., Ltd. (Beijing, China) which is a sandwich enzyme linked immunosorbent assay, was used for the qualitative detection of total (IgG/IgM) antibodies to SARS-CoV-2 spike proteins to the receptor binding domain (RBD) in each processed serum specimen. The assay was used to identify individuals with an adaptive immune response to SARS-CoV-2, indicating recent infection, prior infection, or recent vaccination. First, the plates pre-coated with recombinant receptor-binding domain of the SARS-CoV-2 spike proteins were used to determine the presence of specific SARS-CoV-2 antibodies in the serum samples. Then, recombinant SARS-CoV-2 antigen conjugated to horseradish peroxidase (HRP-Conjugate) enzyme was added to the immunocomplex. Stepwise procedures were used for SARS-CoV-2 antibody detection according to the manufacturer’s protocol [24]. The quality controls used for the study included a negative calibrator (buffer containing newborn calf serum, which is non-reactive to SARS-CoV-2 antibodies) and a positive calibrator (buffer containing monoclonal mouse anti-RBD antibodies in newborn calf serum). These controls were provided by the manufacturer and used as supplied. The test results were considered valid if the following quality control criteria were fulfilled: the absorbance value of the blank well, which contained only Chromogen (Chromogen A contained urease peroxidase, while Chromogen B contained TMB (tetramethylbenzidine)) and stop solution, was <0.080 at 450 nm. Absorbance of the negative calibrator was ≤0.100 at 450 nm after blanking, and the absorbance of the positive calibrator was ≥0.190 at 450 nm after blanking. The absorbance of the antibody captured in the complete immunocomplex was measured using a BioTek^®^ Microplate Reader (Gen 5 3.10), BioTek Instruments, Inc. (Winooski, VT, USA). Results were calculated by relating each specimen’s absorbance value to the cut-off value of the test plate. The cut-off reading was based on a single filter plate reader. First, the absorbance value in the blank well was subtracted from the absorbance value of test samples and controls. The cut-off value was then calculated by adding 0.16 (constant), as stated in the manufacturer’s protocol, to the mean absorbance of the negative control. Each microplate was considered separately when calculating and interpreting the results of the assay, regardless of being processed concurrently. The ratio of the absorbance of the test samples to the cut-off value was used to determine the presence or absence of SARS-CoV-2 total antibodies. Specimens with antibody detection values ˃ 1 were considered reactive, implying the presence of SARS-CoV-2 antibodies. Specimens with antibody detection values < 1 were considered negative, i.e., no serological indication for current or past coronavirus disease. The WANTAI kit has been reported to demonstrate high sensitivity (100%) and specificity (99.7%) [24,25].

### 2.5. Molecular Detection of SARS-CoV-2

Viral RNA was extracted from the swabs using the QIAamp Viral RNA Mini Kit (Qiagen Str, Hilden, Germany) with a vacuum pump extractor. The TIB MOLBIOL (Berlin Germany) LightMix^®^ SARS-CoV-2 E + N UBC Kit and the LightMix^®^ Modular SARS-CoV-2 (COVID-19) RdRP Gene Kit, complemented with the Luna^®^ Universal qPCR Master Mix, were used for the amplification and quantitative detection of SARS-CoV-2. The detection of the virus in a sample was a two-step process, including (1) a screening assay targeting the SARS-CoV-2 E and N genes specific to Cyan500-labeled probes and FAM probes, respectively, and (2) a confirmatory assay targeting the RNA-dependent RNA polymerase gene (RdRP) with a SARS-CoV-2-specific FAM-labeled hydrolysis probe.

A 20 μL RT-qPCR reaction was composed of 5 μL of template RNA, 1 μL of 20X Luna Warmstart RT Enzyme, 10 μL of 2X Luna 1-Step RT-qPCR Master Mix, 0.5 μL of pathogen-specific reagent (PSR) E + N UBC primer-probe, 0.5 μL of PSR Equine Arteritis Virus (EAV) primer-probe, 0.5 μL of EAV control, and 2.5 μL of PCR-grade water. Furthermore, 0.5 μL of PSR (RdRP) was substituted with PSR (E + N UBC primer-probe) for the confirmatory run. Thermocycling conditions used for the assays were set at a reverse transcription step at 55 °C for 10 min and an initial denaturation step at 95 °C for 1 min, followed by 40 cycles of denaturation at 95 °C for 10 s and an extension at 60 °C for 60 s. All RT-qPCR reactions were run on a QuantStudio™ 5 Real-Time PCR Detection System.

Quality controls provided in the TIB MOLBIOL kit were included in all assay plates. The controls included a positive EAV control, spiked as an internal control (ref. 40-0776-96) to detect possible inhibition of PCR, positive controls for the E gene (ref. 40-0776-96) and the RdRP gene (ref. 53-0777-96), and a no-template control (NTC) to detect contamination. The validity of the test was only accepted if the cycle threshold (Ct) value of the E + N and RdRP positive controls was <36 and that of the EAV was <33 and if the NTC did not generate an amplification curve. A sample was considered “conclusively positive” if the Ct values of E + N and RdRP were <36. If the Ct of E + N was <36 but that of the RdRP gene was not observed or >40, the sample was considered “probable” for SARS-CoV-2 virus. A sample was considered “negative” for SARS-CoV-2 if no amplification curve was observed for both the E + N and RdRP genes.

### 2.6. Molecular Detection of Delta and Omicron Variants

The SARS-CoV-2 E-Spike Delta/Omicron TaqMan Typing kit (TIBMOLBIOL: ref. 40-0811-96) (Berlin Germany) was used according to the manufacturer’s protocol to detect the SARS Spike ins214EPE and the SARS Spike del157/158 specific to the Omicron and Delta variants, respectively. A total of 20 μL of RT-qPCR reaction composed of 5 μL of positive SARS-CoV-2 RNA template, 10 μL of 2X Luna 1-Step RT-qPCR Master Mix, 1 μL of PSR (Omicron Delta) primer-probe, which is specific to the detection of the spike proteins of Omicron and Delta variants, 1 μL of 20X Luna Warmstart RT Enzyme, and 3 μL of PCR-grade water was used.

Quality control included a human mRNA sample (Ubiquitin C) provided by the manufacturer as an internal control for the assay. Amplifications (<35) at the FAM and HEX detection channels were accepted as Omicron and Delta positives, respectively. Amplifications detected in the ROX channel were classified as “Not genotyped”, suggesting that they did not match the detection profiles of either Omicron or Delta variants. Thermal cycling conditions used for this assay were the same as those used for the molecular detection of SARS-CoV-2.

### 2.7. Data Analysis

Data collected were cleaned to check for duplications, errors, and completeness using Microsoft Excel 2017. Cleaned data were analyzed using STATA 16 (StataCorp, College Station, TX, USA). Categorical variables, such as age group, sex, place of residence (EA), educational status, vaccination status, occupation, underlying conditions, COVID-19 infection status, travel history, and seropositivity status, were summarized into frequencies and proportions. To assess the precision of our seroprevalence estimate, we calculated a 95% confidence interval (CI) around the point estimate. ArcGISv10.4.1 was used to produce choropleth maps using SARS-CoV-2 prevalence rates. Adherence to COVID-19 preventive measures was determined using nine questions (Table 1). Participants who scored 0–2, 3–6, and 7–9 were classified as low, moderate, and high adherence, respectively. COVID-19 vaccination status among the participants was assessed only for participants aged 18 years and above. This is because only persons in this age category were eligible for COVID-19 vaccines in Ghana at the time of the study.

Both univariate and multivariate binary logistic regression analysis was performed to assess the association between seropositivity status and various participant characteristics. Prior to performing the adjusted logistic regression, a multicollinearity test using the variance inflation factor and a goodness-of-fit test the (Hosmer and Lemeshow model fitness test) were performed to determine the model’s fitness. Robust standard errors were used to adjust for clustering in both the crude and adjusted analyses, with the place of residence as the main clustering variable. To assess the significance of the added predictor variable and its impact on the model’s fit, we performed a likelihood ratio test. All statistical significance was set at *p* ≤ 0.05.

## 3. Results

### 3.1. Background Characteristics of the Study Participants

Out of the 1027 participants recruited, 966 (94.1%) were from urban settlements. Regarding age distribution, 390 (38.0%) were <20 years old. More than half, 580 (56.5%) of the participants, had secondary or higher educational levels. Less than one-third, 296 (29.0%), were vaccinated against SARS-CoV-2. The vast majority of participants, 1000 (97.4%), stated they had no underlying conditions. Regarding adherence to recommended SARS-CoV-2 prevention and control measures, 393 (38.3%) were identified to have low adherence, while 293 (28.53%) reported no adherence. Collectively, the majority of suboptimal adherence to COVID-19 prevention protocols in the study population was attributed to these groups, which accounted for 686 (66.8%) (Table 1).

### 3.2. SARS-CoV-2 Seroprevalence, Infection, and Circulating Variants

In our study sample, SARS-CoV-2 seropositivity was 86.8% [95% CI: 84.53–88.77]. Among the 1027 study participants, 105 (10.2%) showed RT-PCR positive status at the time of sampling. More than half of RT-PCR positive study participants were unvaccinated (67/105, 64%). A comparison of infection status by vaccination status showed that more unvaccinated participants tested positive compared to vaccinated participants. The Omicron variant accounted for 42.86% of the circulating variants (Table 2), while the highest number of SARS-CoV-2 positive cases was recorded in the Ashiedu–Keteke district (17.1%) (Figure 1).

### 3.3. Factors Associated with SARS-CoV-2 Seropositive Status in the Study Population

In all, 844/966 (87.4%) of participants from the urban areas were seropositive, compared to 47/61 (77.1%) in the rural settlement. The median age of seropositive participants was 29 years (IQR:16–44). More employed participants 447/491 (91.0%) were seropositive than unemployed persons (444/536, 82.8%). The univariate logistic regression showed that the type of settlement, age, employment status, and vaccination status showed strong evidence of an association with serostatus.

Study participants from urban settlements had more than two-fold increased odds of seropositivity compared to those from rural settlements (cOR = 2.06, 95% CI: 1.07–3.76, *p* = 0.024). Furthermore, an increase in age stratum was associated with 3% increased odds of seropositivity among participants (cOR = 1.03, 95% CI: 1.02–1.05, *p* < 0.001). The 60+ year group showed six times increased odds of seropositivity (cOR = 6.05, 95% CI: 2.44–20.2, *p* < 0.001). Vaccinated participants had 10.5 times increased odds of seropositivity than those not vaccinated (cOR = 10.5, 95% CI: 4.97–26.9, *p* < 0.001) (Table 3). The logistic regressions models’ goodness of fit determined that the removal of the categorical variables age, contact with a person with flu-like symptoms, and adherence level did not significantly improve the fit of the model compared to the model with these predictors (LR χ^2^ = 7.52; *p*-value = 0.4816). The multivariate logistic regression results showed that unemployed participants, secondary+-level participants, and vaccination status had a statistically significant association with serostatus after adjustments for other factors. The adjusted odds of seropositivity were 2.10 times higher for those in secondary+ level than those that have never attended school (aOR = 2.10, 95% CI: 1.12–3.83, *p* = 0.018). Vaccinated participants had 8.53 greater odds of seropositivity than non-vaccinated individuals (aOR = 8.53, 95% CI: 3.87–22.6, *p* < 0.001). The adjusted odds of seropositivity were 0.52 times lower for unemployed participants than the employed ones.

### 3.4. Factors Associated with SARS-CoV-2 Seropositive Status Among Non-Vaccinated Participants

Educational level and employment status were identified to be significantly associated with seropositivity. Individuals with secondary or higher education had significantly higher odds of being seropositive compared to those with no formal education (cOR = 2.09, 95% CI: 1.13–3.77, *p* = 0.016). After adjusting for potential confounders, the odds of seropositivity remained significantly higher (aOR = 2.23, 95% CI: 1.17–4.17, *p* = 0.013) among those with secondary or higher education compared to those with no formal education. Additionally, unemployed individuals had significantly lower odds of being seropositive compared to those who were employed (cOR = 0.58, 95% CI: 0.38–0.86, *p* = 0.009). The adjusted odds of seropositivity were 0.41 times lower for unemployed participants (aOR = 0.41, 95% CI: 0.21–0.80, *p* = 0.009) than their employed counterparts (Table 4).

## 4. Discussion

This study was conducted to complement the epidemiological data on COVID-19 in Ghana, especially during the Omicron infection wave and after several months of the vaccination initiative against the disease. This study was conducted in Accra, which was established as one of the cities with a high burden of SARS-CoV-2 [16]. We found an overall SARS-CoV-2 seroprevalence of more than 80% in the study population. The high prevalence of SARS-CoV-2 total antibodies detected in our study was higher than the national average reported in Ghana [26]. While our findings are not directly comparable to other studies in Ghana, they align with trends of high seroprevalence observed in some African countries, and they are similar to those of other reports from Ghana and other African countries [27]. Two reasons could explain the high seropositivity: (1) high exposure rates to the SARS-CoV-2 virus and/or (2) high vaccine uptake. We found a significant association between seroprevalence and vaccination after adjustments for other factors (aOR = 8.53, 95% CI: 3.87–22.6, *p* < 0.001) (Table 3). The comparison of variables associated with seropositivity between the total population (both vaccinated and non-vaccinated) and only the non-vaccinated individuals suggests that area of residence, age group, and sex were significant predictors of seropositivity in the overall population but lost significance when the analysis was restricted to non-vaccinated individuals. This shows that vaccination may have influenced the association of certain variables and seropositivity (Table 4). Also, we found a significant difference between the relative absorbance values for vaccinated and unvaccinated participants (Appendix A). The WANTAI ELISA assay is unable to distinguish antibodies resulting from exposure, infection, and vaccination; therefore, we could not explore these associations further. This was the main limitation of our study. Older age groups had higher seropositivity rates in our study. This is consistent with earlier research reporting increased seropositivity rates among older people [28].

The high seropositivity rates observed in this study occurred after mass vaccination and two epidemic peaks (Alpha and Delta waves). Test positivity reported in this study was 10%, which was comparable to the 11.8% reported by the Ghana Health Service (GHS) between March 2020 and December 2021 using routine surveillance [20]. However, when making this comparison, it should be noted that the study population included a higher proportion of asymptomatic cases compared to routine surveillance, which mostly captures symptomatic cases.

The SARS-CoV-2 variants of concern have impacted virus transmissibility and pathogenicity as well as vaccine effectiveness [29]. Evidence from countries with documented transmission and high levels of population immunity suggests that the Omicron variant has a growth advantage over the Delta variant [30,31]. It remains uncertain to what extent the rapid growth rate can be attributed to immune evasion, intrinsic increased transmissibility, or a combination of both. However, recent evidence confirms that the Omicron variant emerged because of its ability to evade pre-existing human immunity [32]. A high prevalence of Omicron was expected because the study was conducted during a period when this variant wave was occurring. However, almost half (47%) of the infected participants had variants classified as “Not genotyped”. This suggests that some circulating variants were not specifically identified and that new variants do not rapidly outcompete existing ones, allowing them to persist long enough for potential recombination events. However, our study did not further characterize the detected variants classified as “Not genotyped” in the study population or investigate how they were associated with the parameters investigated in this study. The presence of Omicron was nonetheless high among the study population, supporting reports of its high transmissibility [33]. This information also suggests the need for enhanced genomic surveillance efforts and encourages vaccination in Ghana.

On average, 30–40% of the population responded that they adhered to at least one of the established WHO COVID-19 infection prevention guidelines. It is important to note that none of the participants had knowledge of their infection status prior to the visit to their households for inclusion in the study. As such, they were not expected to be practicing self-isolation, for example, as they had not been confirmed positive for the virus. It was observed that a significant majority responded that they did not practice other general preventative measures, such as frequent handwashing, social distancing, and avoiding non-essential social contact. It was, however, encouraging to note that most people who tested positive for the virus wore face masks. For a population that is largely asymptomatic for the disease, wearing of face masks is by far one of the best ways to avoid community spread of the virus.

## 5. Conclusions

This study represents a snapshot of SARS-CoV-2 seroprevalence, active infection, and variant distribution in Greater Accra during the Delta–Omicron wave. The seropositivity observed (86%) suggests high exposure rates to the virus and/or vaccine-induced immunity. Concurrently, 10% of the participants tested positive for SARS-CoV-2, with Omicron emerging as the predominant variant. Our findings highlight the impact of vaccination, as vaccinated individuals had significantly higher odds of seropositivity compared to unvaccinated individuals. Despite the high immunity levels, low adherence to COVID-19 prevention protocols may have facilitated extensive community transmission. Nonetheless, this study underscores the necessity for sustained vaccination efforts to mitigate the transmission and impact of future SARS-CoV-2 waves. Further longitudinal studies incorporating genomics surveillance are needed to fully understand variant evolution and transmission patterns in Ghana.

## Figures and Tables

**Figure 1 viruses-17-00487-f001:**
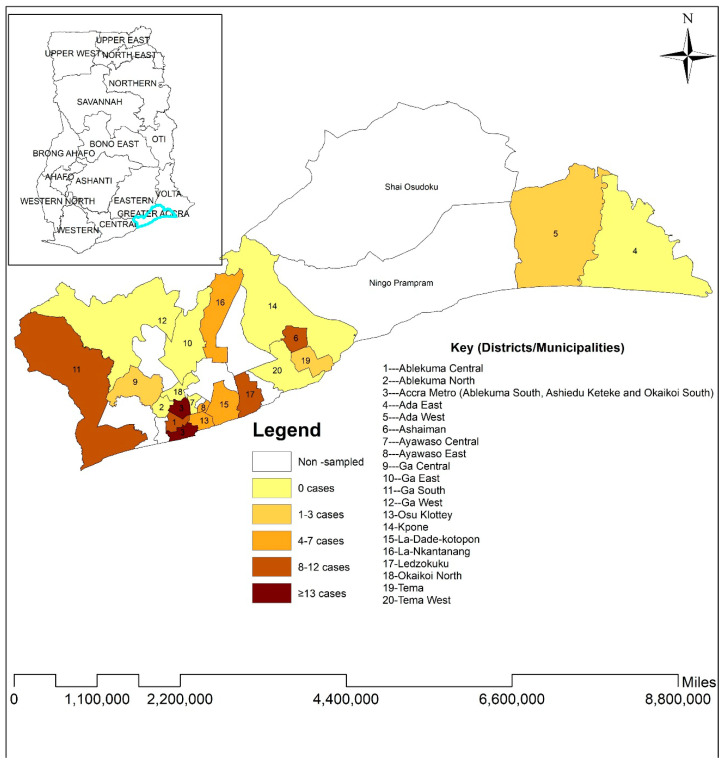
Distribution of SARS-CoV-2 infection using RT-PCR in the Greater Accra Region. Prevalence map shows the total number of individuals with RT-PCR positive status in 22 districts/municipalities.

**Table 1 viruses-17-00487-t001:** Sociodemographic characteristics of study participants.

Characteristic	Frequency (N = 1027)	Percentage (%)
Area		
Rural	61	5.94%
Urban	966	94.06%
Age group		
<20	390	37.97%
20–39	353	34.37%
40–59	180	17.53%
60+	104	10.13%
Sex		
Female	575	55.99%
Male	452	44.01%
Educational level		
Never attended school	94	9.15%
Primary	353	34.37%
Secondary+	580	56.48%
Employment status		
Employed	491	47.81%
Unemployed	536	52.19%
Vaccination status		
No	731	71.18%
Yes	296	28.82%
Pre-existing medical conditions		
No	1000	97.37%
Yes	27	2.63%
Have you had contact with anyone with flu-like symptoms in the last 7 days?		
No	955	92.99%
Unknown	24	2.34%
Yes	48	4.67%
Adherence to COVID-19 protocols ^ɸ^		
High	183	17.82%
Low	393	38.27%
Moderate	158	15.38%
No adherence	293	28.53%

^ɸ^ Adherence to the COVID-19 protocols is categorized as follows: high adherence (compliance with 7–9 measures), moderate adherence (compliance with 3–6 measures), low adherence (compliance with 1–2 measures), and no adherence (non-compliance with any measures).

**Table 2 viruses-17-00487-t002:** SARS-CoV-2 seroprevalence, infection rate, and circulating variants among the study participants.

Characteristic	Frequency (N)	Percentage (%)	Vaccination Status		*p*-Value
			No, N = 731	Yes, N = 296	
SARS-CoV-2 infection					0.079
Negative	922	89.78%	664 (72.0%)	258 (28.0%)	
Positive	105	10.22%	67 (63.8%)	38 (36.2%)	
SARS-CoV-2 serostatus					<0.001
Negative	136	13.24%	130 (95.6%)	6 (4.4%)	
Positive	891	86.76%	601 (67.5%)	290 (32.5%)	
Circulating variants					0.042
Omicron	45	42.86%	29 (64.4%)	16 (35.6%)	
Delta	9	8.57%	9 (100.0%)	0 (0.0%)	
Not genotyped	51	48.47%	29 (56.9%)	22 (43.1%)	

**Table 3 viruses-17-00487-t003:** Factors associated with SARS-CoV-2 serostatus among the study participants.

Characteristic	Serostatus		Univariate Regression		Multivariate Regression	
	Negative, N = 136	Positive, N = 891	cOR (95% CI) ^1^	*p*-Value	aOR (95% CI) ^2^	*p*-Value
Area						
Rural	14 (23.0%)	47 (77.0%)	—		—	
Urban	122 (12.6%)	844 (87.4%)	2.06 (1.07–3.76)	0.024	1.88 (0.93–3.60)	0.065
Age group						
<20	76 (19.5%)	314 (80.5%)	—		—	
20–39	39 (11.0%)	314 (89.0%)	1.95 (1.29–2.98)	0.002	0.75 (0.40–1.44)	0.4
40–59	17 (9.4%)	163 (90.6%)	2.32 (1.36–4.18)	0.003	0.68 (0.30–1.59)	0.4
60+	4 (3.8%)	100 (96.2%)	6.05 (2.44–20.2)	<0.001	2.26 (0.81–8.14)	0.2
Sex						
Female	75 (13.0%)	500 (87.0%)	—		—	
Male	61 (13.5%)	391 (86.5%)	0.96 (0.67–1.39)	0.8	0.98 (0.67–1.43)	0.9
Educational level						
Never attended school	19 (20.2%)	75 (79.8%)	—		—	
Primary	58 (16.4%)	295 (83.6%)	1.29 (0.71–2.26)	0.4	1.68 (0.89–3.11)	0.10
Secondary+	59 (10.2%)	521 (89.8%)	2.24 (1.24–3.90)	0.006	2.10 (1.12–3.83)	0.018
Employment status						
Employed	44 (9.0%)	447 (91.0%)	—		—	
Unemployed	92 (17.2%)	444 (82.8%)	0.48 (0.32–0.69)	<0.001	0.52 (0.28–0.98)	0.040
Vaccination status						
No	130 (17.8%)	601 (82.2%)	—		—	
Yes	6 (2.0%)	290 (98.0%)	10.5 (4.97–26.9)	<0.001	8.53 (3.87–22.6)	<0.001
Pre-existing medical conditions						
No	135 (13.5%)	865 (86.5%)	—		—	
Yes	1 (3.7%)	26 (96.3%)	4.06 (0.85–72.7)	0.2	1.90 (0.34–35.7)	0.5
Have you had contact with anyone with flu-like symptoms in the last 7 days?						
No	130 (13.6%)	825 (86.4%)	—		—	
Unknown	2 (8.3%)	22 (91.7%)	1.73 (0.50–10.9)	0.5	1.97 (0.53–12.8)	0.4
Yes	4 (8.3%)	44 (91.7%)	1.73 (0.69–5.83)	0.3	1.63 (0.61–5.70)	0.4
Adherence to COVID-19 protocols						
High	24 (13.1%)	159 (86.9%)	—		—	
Low	56 (14.2%)	337 (85.8%)	0.91 (0.54–1.50)	0.7	1.04 (0.58–1.81)	0.9
Moderate	19 (12.0%)	139 (88.0%)	1.10 (0.58–2.12)	0.8	1.11 (0.55–2.24)	0.8
No adherence	37 (12.6%)	256 (87.4%)	1.04 (0.60–1.80)	0.9	1.64 (0.89–2.98)	0.11

^1^ cOR = Crude Odds Ratio, ^2^ aOR = Adjusted Odds Ratio.

**Table 4 viruses-17-00487-t004:** Factors associated with SARS-CoV-2 serostatus among non-vaccinated participants.

Characteristic	Serostatus		Univariate Regression		Multivariate Regression	
	Negative, N = 130	Positive, N = 601	cOR ^1^ (95% CI)	*p*-Value	aOR ^2^ (95% CI)	*p*-Value
Area						
Rural	14 (23.3%)	46 (76.7%)	—		—	
Urban	116 (17.3%)	555 (82.7%)	1.46 (0.75–2.67)	0.2	1.90 (0.94–3.66)	0.063
Age group						
<20	76 (20.4%)	296 (79.6%)	—		—	
20–39	34 (15.0%)	193 (85.0%)	1.46 (0.94–2.29)	0.10	0.69 (0.36–1.35)	0.3
40–59	16 (17.8%)	74 (82.2%)	1.19 (0.67–2.22)	0.6	0.49 (0.21–1.22)	0.12
60+	4 (9.5%)	38 (90.5%)	2.44 (0.94–8.32)	0.10	1.75 (0.61–6.40)	0.3
Sex						
Female	69 (17.2%)	333 (82.8%)	—		—	
Male	61 (18.5%)	268 (81.5%)	0.91 (0.62–1.33)	0.6	0.90 (0.61–1.34)	0.6
Educational level						
Never attended school	19 (27.1%)	51 (72.9%)	—		—	
Primary	57 (18.8%)	247 (81.3%)	1.61 (0.87–2.91)	0.12	1.78 (0.93–3.32)	0.075
Secondary+	54 (15.1%)	303 (84.9%)	2.09 (1.13–3.77)	0.016	2.23 (1.17–4.17)	0.013
Employment status						
Employed	38 (13.1%)	251 (86.9%)	—		—	
Unemployed	92 (20.8%)	350 (79.2%)	0.58 (0.38–0.86)	0.009	0.41 (0.21–0.80)	0.009
Pre-existing medical conditions						
No	129 (18.0%)	587 (82.0%)	—		—	
Yes	1 (6.7%)	14 (93.3%)	3.08 (0.61–56.0)	0.3	1.81 (0.31–34.4)	0.6
Have you had contact with anyone with flu-like symptoms in the last 7 days?						
No	124 (18.1%)	561 (81.9%)	—		—	
Unknown	2 (10.0%)	18 (90.0%)	1.99 (0.56–12.6)	0.4	1.92 (0.51–12.6)	0.4
Yes	4 (15.4%)	22 (84.6%)	1.22 (0.46–4.21)	0.7	1.52 (0.55–5.41)	0.5
Adherence to COVID-19 protocols						
High	23 (19.3%)	96 (80.7%)	—		—	
Low	54 (20.7%)	207 (79.3%)	0.92 (0.53–1.57)	0.8	1.02 (0.56–1.81)	>0.9
Moderate	17 (16.2%)	88 (83.8%)	1.24 (0.62–2.50)	0.5	1.20 (0.58–2.52)	0.6
No adherence	36 (14.6%)	210 (85.4%)	1.40 (0.78–2.47)	0.3	1.69 (0.90–3.12)	0.10

^1^ cOR = Crude Odds Ratio, ^2^ aOR = Adjusted Odds Ratio.

## Data Availability

All data generated or analyzed during this study are included in this published article and its Appendix A.

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
