# Peer review of "Evaluation of SARS-CoV-2 Seroprevalence and Variant Distribution During the Delta–Omicron Transmission Waves in Greater Accra, Ghana, 2021"

_viruses, 2025, doi:10.3390/v17040487_

Round 1
Reviewer 1 Report
Comments and Suggestions for Authors
General comments.
Molecular typing of SARS-CoV-2 isolates from patients in Greater Accra, Ghana is based on reverse trascription with real time PCR with commercial kits to identify two variants Delta and Omicron whereas other half (48,57% according to Table 2) remained unkown. Sequencing of the PCR products with phylogenetic analysis are highly desirable.
SARS-CoV-2 Omicron appeared to be the variant of concern (VOC) on 26 Nov 2021. So until December 2021 Omicron could not become the prevailing variant of the SARS-CoV-2 in Africa.
Gradation of adherence to COVID-19 protocols (Table 1) (high, moderate and low) is not defined.
Origin of antigen(s) including SARS-CoV-2 variant and strain, isolation and purification methods should be described more carefully since amino acid substitutions are known in the SARS-CoV-2 RBD domain.
Research with clinical samples must include informed consent forms from adults, assent form for children and adolescents as well as approval from the Ethics Committee with corresponding numbers.
Minor comments are in the attached file.

Both grammar and spelling should be carefuuly checked throughout the manuscript.
Author Response
Dear Reviewer,
Thank you for the time spent to review our manuscript. Please see below our responses to your comments.
- Comment 1: Molecular typing of SARS-CoV-2 isolates from patients in Greater Accra, Ghana is based on reverse transcription with real time PCR with commercial kits to identify two variants Delta and Omicron whereas other half (48,57% according to Table 2) remained unknown. Sequencing of the PCR products with phylogenetic analysis are highly desirable.
Response 1: Thank you for pointing this out. We agree with this comment and appreciate your suggestion regarding sequencing and phylogenetics analysis. Unfortunately, due to financial constraints, we were unable to sequence the positive samples to determine the unknown variant. This remains a limitation of our study, and we recognize that sequencing efforts would have provided a more comprehensive understanding of the circulating SARS-CoV-2 in Greater Accra. We hope that future studies with additional funding will address these gaps.
- Comment 2: SARS-CoV-2 Omicron appeared to be the variant of concern (VOC) on 26 Nov 2021. So, until December 2021 Omicron could not become the prevailing variant of the SARS-CoV-2 in Africa.
Response 2: We appreciate your comment and recognize the importance of accurately representing the timeline of SARS-CoV-2 Omicron emergence. While the World Health Organization designated Omicron as a Variant of Concern (VOC) on November 26 2021, multiple reports and genomic surveillance data suggest that Omicron was already in several African countries before its official classification (https://www.nature.com/articles/s41392-022-00997-x) (https://pmc.ncbi.nlm.nih.gov/articles/PMC9629902/). By early December 2021, Omicron had rapidly outcompeted other variants due to its high transmissibility. For instance, South Africa reported a sharp increase in Omicron cases as early as late November 2021, with the variant becoming dominant by the beginning of December. Other African countries including Botswana and Ghana, also documented Omicron’s rapid spread during this period(https://www.graphic.com.gh/news/general-news/covid-19-ghana-records-cases-of-omicron-variant-at-airport.html). (Emergence and spread of the SARS-CoV-2 omicron (BA.1) variant across Africa: an observational study - The Lancet Global Health)
- Comment 3: Gradation of adherence to COVID-19 protocols (Table 1) (high, moderate and low) is not defined.
Response 3: Thank you very much for your question reviewer 1. The gradation of adherence to the COVID-19 protocols is defined in the Data Analysis section (lines 232-233). We have also added a footnote to (Table 1) to clarify these definitions for easier reference.
High Adherence: Compliance with seven to nine measures.
Moderate Adherence: Compliance with three to six measures.
Low Adherence: Compliance with one or two measures.
No Adherence: Non-compliance with any measures.
The nine inquiries used to assess adherence were: 1) Increased frequency and duration of handwashing 2) Refraining from nonessential social contact or interactions 3) Practicing social distancing 4) Avoiding visits to places of worship, cafes, and restaurants whenever feasible
5) Engaging in self-isolation at home 6) Using hand sanitizing gel more frequently 7) Avoiding public transportation whenever possible 8) Working remotely or increasing the frequency of remote work 9) Wearing a mask.
- Comment 4: Origin of antigen(s) including SARS-CoV-2 variants and strain, isolation and purification methods should be described more carefully since amino acid substitutions are known in the SARS-CoV-2 RBD domain.
Response 4: Thank you very much for your comment reviewer. We appreciate your guidance in making our paper relevant. In our study, we used the WANTAI SARS-CoV-2 Ab ELISA kit (WS-1096), which detects antibodies against the receptor-binding domain (RBD) of the spike protein using a recombinant antigen. This recombinant protein was derived from the Wuhan-Hu-1 strain. It is important to clarify that our laboratory did not isolate or purify the RBD. Instead, we relied on this validated commercial kit, which has been widely used in serological studies to detect antibodies against this specific domain. The WANTAI kit was evaluated and was recommended for use by the World Health Organization for its high sensitivity and specificity in detecting SARS-CoV-2 antibodies (https://pmc.ncbi.nlm.nih.gov/articles/PMC8711170/). Participating countries in the WHO UNITY studies all utilized this kit. Additionally, while amino acid substitutions in RBD domain are known to occur across variants, the recombinant antigen in this assay at the time of our study we believe was designed to retain key conserved epitopes, ensuring broad detection capability.
- Comment 5: What chromogenic substrate was used together with hydrogen peroxide?
Response 5: Thank you very much for your comment. In the WANTAI kit , we used a two-component chromogen system. Chromogen A contains Urea peroxidase. Chromogen B contains TMB (Tetramethylbenzidine). The chromogenic substrate that reacts with Urea peroxide in the reaction is TMB (Tetramethylbenzidine).
- Comment 6: Units? Optical units at 450 nm or something else?
Response 6: Thank you very much for your comment. The values given (A/C.O. <1 or ≥ 1) represent the ratio of samples optical density (OD) absorbance measured at 450 nm to the cut-off value (C.O.), which is also measured in OD by the assay. Since both values are in the same unit (OD), their ratio is dimensionless and indicates whether the samples absorbance is below or above the cut-off threshold.
- Comment 7: Research with clinical samples must include informed consent forms from adults, assent form for children and adolescents as well as approval from the Ethics Committee with corresponding numbers.
Response 7: Thank you for your comment. Ethical approval was obtained and informed consent forms was also obtained from all our research participants, including assent for minors. All of the above have been included in the paper. Kindly refer to our Informed consent Statement and Institutional Review Board Statement. You can also find below:
“Informed Consent Statement: Informed consent was obtained from all subjects and/or their legal guardians (for subjects aged below 18 years). All consenting participants agreed to future use of their samples. Both verbal and written consent was obtained from all participants in the presence of a witness and from the parent/guardian of minors; this was done to ensure that the study was thoroughly explained to the participants and their parents/guardians.
Institutional Review Board Statement: Procedures in this study conform with the Ghana Public Health Act, 2012 (Act 851) and the Data Protection Act, 2012. Ethical approval (NMIMR-IRB CPN 075/19-20) was obtained from the Institutional Review Board (IRB) of NMIMR, University of Ghana.

Reviewer 2 Report
Comments and Suggestions for Authors
The paper by Lomotey and colleagues is an epidemiological report about SARS-COV-2 seroprevalence in Ghana in 2021. Despite these data being very old, and currently of no substantial utility, it may be interesting to bring them back to life for improving some elements of understanding about that specific phase of the pandemics.
The introduction has some repetitions, for example whend dealing with local Ghanaian measures, but the main limit is that it relies mostly on outdated data, and it is not clear if this is intentional or not, especially because
the aim is focused on a 'past' time frame (and the reason for this choice is not actually clarified). References are often from 'grey' literature, or from outdated papers, or from commentaries, letters, webpages, while I think that a more
accurated work of literature search would bring out more relevant references.
The methods section seem to be correctly informative, except for minor points, but it provides me with a doubt that will be also extended to the results section: To which extent the presented work is actually original and to which extent is it a
included in the already published paper about the nation-wide investigation (ref 26)? If I understand correctly, the presented cases are a subset of those sampled for the previously published work, so the sampling and the serological analysis shown here as original had actually
been already performed in that occasion, as well as the viral detection. The viral genotyping, at difference, appears to be completely original. I think there should be a clear notification of this in the methods section, ans well as in the results, to cearly
state to the reader which of the methods/results presented are original and which ones are republished or repurposed for the occasion.
Of course, the same concepts applies, and to an even greater extent, to the results section. I really would like to urge the authors to clearly identify also in every results' paragraph, what is original of this specific publication and what is a re-analysis
of a subset of the wider one from ref 26 (or from others, in case).
Also, about results, along with tables and statistical comparisons, it could also be recommended to provide rawer data for the seroprevalence tests (a table with numerical data and the provided S1 could be added to main text).
Finally, I think it would be possible, since the active SARS-COV-2 infection is identified in a sufficiently high number of cases, to try to seek for associations with socio-demographic variants. Also, as stated in the minors, please add at least the association with vaccination status, which is commented in the text, in the relavant tables
The discussion deals with the very striking seroprevalence found in that studio, failing to reconnect it with similar findings (see minor points), and pointing out a possible explanation: the assay used, which can't distinguish IgG and IgM, nor neutralizing vs non-neutralizing antibodies, thus making it impossible to determine the relative component due to vaccine or natural infection.
This, as proposed elesewhere, could be somehow mitigated by narrowing the selection to non-vaccinated subjects. Moreover,in this reviewer's humble opinion, the single method used for Ab analysis appears to be a bad strategy. At least a smaller validation sub-set tested with an alternative method would have provided maybe more reliable results.
My fear is that the only method used could have some implicit limitations, for example a too-low cut-off, bringing to a neat excess of positivities.
In the impossibility of repeating the test, I would suggest to try to perform a second data evaluation and subsequent statistical analysis by using a higher value (e.g., 2) and see if something changes significantly....
In summary, I would encourage the aithors to try to make a general check on the notified weakpoints of the paper, and try to performed some methodological and/or analyitical improvement. For giving them time for this, I'll request a major revision.
Some minor points:
in page 1 (introduction): - the number of worldwide deaths for COVID-19 seems to be neatly underextimated, and indeed the updated reference 1 page reports a far larger measure. Please update.
in page 2 (introduction): - the number of active cases are not relevant',since they're not cumulative . Also, these data are old and should be updated. The link to the database should not be in the text but in the references
- in the aims sections, please remind the readers which timeframe corresponds to the mentioned 3rd wave.Furthermore, the sampling dates and also the paper's title seem to suggest that the observation has been actually performed during the 4th wave (or do I miss something?. Please clarify.
in page 3 (metohds): - in line 1, it is not so clear what do this dates refer to. The nation-level study (ref 26) is reporting data from a much longer timeframe.
- paragraph 2.2 refers to population, but it actually only deals with geographical informations, so please change the title. Alternatively, merge with paragraphs 2.3 and 2.4 (too short and not sufficiently informative to be considered for single paragraphs)
- in par 2.6, why is the ELISA assay called "WHO"?
in page 4 (methods): - still in par 2.6, the manufacturer's protocol is actually linked to a preprint paper. Please add a link to the actual manufacturer protocol or to another paper describing it
Also, in the final sentence, please specify that these numbers come from a specific paper (ref 29, since I don't even consider ref 28), with a specific design, and are not necessarily the manufacturer's declared performances
in page 5 (methods): - in par 2.8, please provide a reference for the product manufacturer's protocol. Please explain more clearly the concepts behind the assay, especially the role of the rox reporter.The main doubt arises from the fact that ROX is associated with a kind of 'aspecific'
positivity, which in the final results (see relevant section), accounts for the majority of cases, which don't seem to be so expatable.
in page 6 (results): - in table 1, there's a note '1' near the number of participants . What does this refer to?
- in par 3.1, is it possible to know, or to extimate, which vaccine was/had been in use up to that period in that area?
- still in par 3.1, it would be interesting to add, if available, the information about 'previously reported COVID-19'.
- still in par 3.1, the proportion of people declaring no adherence to restrictions is also interesting, and could also be highlighted (as much as the total no+low adherence, which forms a wide majority of the population)
- in par 3.2, please introduce reference to tab2 after seroprevalence and RT-PCR status.
- stillin par 3.2, data about association between infection and vaccination status are presented, but are not present in any table. Please add.
in page 7 (results): - in par 3.2, line 2, the proportion of omicron variant is stated as 42.86%, but actually the majority of the typed positivities are listed as 'other' variants. Now, this poses a serious doubt over the test interpretation, since it is possible that some of these
'others' could be sub-variants of omicron, or minor types still circulating, but it looks unlikely that they could represent the majority of the positive cases. It looks like more probable that the actual omicron and delta proportion is therefore underxtimated.
Maybe, testing with a different assay would change the proportion dramatically, but probably this is no more possible. So, my suggestion is to go more in detail into understanding the technical issue beyond the assay's odd performance, and understanding if, maybe, only
the omicron+delta cases correctly genotype could be used for the calculation of the relative proportions, considering the others as 'not genotyped' rather than 'other genotypes'.
in page 9 (results): - in par 3.3, as shown also in detail in table 3, the role of vaccination status towards seropositivity rate is obviously relevant. In all other variables taken into account, there could be a bias due to that, since it is possible that vaccination status is unbalanced in specific
age classes, type of settlement, and more in general in all social classification groups. So, it would be more informative to add a stratification of the other variables with respect ot vaccination status, or to take into consideration, for relevant variable, also to show results from 'vaccine-free' population.
I understand that seroprevalence is a cumulative concept, but the association with other factors has very different interpretation wether the seroprevalence is due to natural infection or vaccines.
in page 9 (discussion): - the high seroprevalence is said to be confirmed in other African countries, but ref 17,30 provides very different results.
- the strong association between vaccination and seroprevalence quite obvious, and it is clear that seroprevalence studies performed during the vaccination era are completely different from the first ones, based entirely on natural immunity
- the strong correlation with age in ref 31 is found on vaccinated subjects. On non vaccinated it is shown to be the opposite. In this study, the large majority of seropositives are non vaccinated, so the trend is very different than in ref 31.
in page 10 (discussion): - The first period (starting actually from the end of page 9), describes a significant association with pre-existing conditions, but this is excluded in the results. Please check. By the way, the total number of subjects with pre-existing conditions would anyway be too low to get relevant associations.
- the proportion of acute infections is really high, probably much higher than that reported in the same time period in americas and europe. The cited papers (ref 32-34) don't show any similar, nor higher, data. Please check.Also the GHS bullettins from 2021, seem to show incidence rates below 1%.
In general terms, a 10% positive cases in a random population screening are something quite striking.
- reference 36 is completely off-topic
in the references section: - please update the link for reference 1
- please check for correct link for reference 2
- despite being relevant publications, both ref 3 and 4 are very old; a much more recent selection should be associated to the specific sentence in the text.
- same as above for refs 5 and 6. Coming from the very first year of infection, they're too old to fit the sentence (which deals with 2-years period)
- ref 10 is actually the announcement of an upcoming paper which under review in 2022. Please add current paper link of select a different source
- ref 18 lacks journal's details. Also, author's name is miswritten
- ref 20 has a wrong link. Moreover, please describe GHS acronym
- for ref 21, please provide a more specific link
- for ref 24, lease provide link or further details for the paper
- for ref 25, link for publication is not working or mispelled. Please correct or update
- reference 28 is from a preprint with no peer-review from 2020. please update or select another source
- ref 34 is from a continuosly updated page. Please provide a specific link for the bullettin chosen for comparison in the relevant period.
- ref 37 title's is mispelled and date is wrong
Author Response
Dear Reviewer,
Thank you for the time spent to review our manuscript. Please see below our responses to your comments.
- Comment 1: (introduction): The number of worldwide deaths for COVID-19 seems to be neatly underestimated, and indeed the updated reference 1 page reports a far larger measure. Please update.
Response 1: Thank you very much for your comment. We have updated the manuscript to reflect a more recent data. Please refer to line 2 of the introduction for the revised figures and updated reference.
- Comment 2: (introduction): - the number of active cases are not relevant, since they're not cumulative . Also, these data are old and should be updated. The link to the database should not be in the text but in the references.
Response 2: Thank you for your comment. We have effected changes to the comments. Lines 34-35 of the Introduction have been revised. We have also moved the database link to the references.
- Comment 3: In the aims sections, please remind the readers which timeframe corresponds to the mentioned 3rd wave. Furthermore, the sampling dates and also the paper's title seem to suggest that the observation has been actually performed during the 4th wave (or do I miss something? Please clarify
Response 3: Thank you very much for drawing our attention to your comment. Manuscript has been duly revised. Kindly refer to line 64 of the Introduction for the revision.
- Comment 4: In page 3 (methods): - in line 1, it is not so clear what do these dates refer to. The nation-level study (ref 26) is reporting data from a much longer timeframe.
Response 4: Thank you very much for your comment. Kindly find revised lines 82-83 in the methods section to clarify the timeframe. The dates refer specifically to the period during which this study was conducted and do not correspond to the longer timeframe reported in the national-level study( reference 26, now Reference 25). Additionally, the data presented in this study were not included in the national-level study.
- Comment 5: Paragraph 2.2 refers to population, but it actually only deals with geographical informations, so please change the title. Alternatively, merge with paragraphs 2.3 and 2.4 (too short and not sufficiently informative to be considered for single paragraphs)
Response 5: Thank you very much for your comment. We have merged paragraphs 2.2, 2.3, and 2.4 into a single section to improve coherence and readability. Additionally, we have revised the section title to better reflect its content.
- Comment 6: in par 2.6, why is the ELISA assay called "WHO"?
Response 6: Thank you very much for pointing this out. We referred to it as WHO indicating the World Health Organization because it was the test kit recommended for al countries participating in the UNITY studies at the time. However, we have revised the line of the paragraph 2.6. for clarity.
- Comment 7: In page 4 (methods):- still in par 2.6, the manufacturer's protocol is actually linked to a preprint paper. Please add a link to the actual manufacturer protocol or to another paper describing it.
Response 7: Thank you for your comment. We have updated the reference in Paragraph 2.6 to include a link to the actual manufacturer’s protocol. Kindly refer to reference 28.
- Comment 8: Also, in the final sentence, please specify that these numbers come from a specific paper (ref 29, since I don't even consider ref 28), with a specific design, and are not necessarily the manufacturer's declared performances.
Response 8: Thank you very much for your comment. We have clarified in the final sentence that the reported numbers are derived from (Reference 29, now Reference 28), which is from an independent evaluation. Additionally, (previously reference 28, now reference 27) now correctly captures the manufacturer’s protocol.
- Comment 9: In page 5 (methods):- in par 2.8, please provide a reference for the product manufacturer's protocol. Please explain more clearly the concepts behind the assay, especially the role of the rox reporter. The main doubt arises from the fact that ROX is associated with a kind of 'aspecific' positivity, which in the final results (see relevant section), accounts for the majority of cases, which don't seem to be so expatable.
Response 9: Thank you for your comment. This assay is specifically designed for the detection and differentiation of the Delta and Omicron variants of SARS-CoV-2. It targets specific mutations within the spike protein that are characteristic of each variant. The probe specific to the Omicron variant is labelled with FAM dye and the probe specific for the Delta variant is labelled with HEX dye. We acknowledge that ROX is a passive reference dye used to normalize fluorescence signal differences due to differences in optical path length. Again, we agree on “aspecific positivity”. Amplification at the ROX channel were considered as “other variants”. It is possible that the “aspecific” positivity observed in the ROX channel could be attributed to the detection of SARS-CoV-2 variants other than Delta or Omicron, or from non-specific signals. Ideally, sequencing would have been performed to confirm the identities of these variants, however, this was not feasible due to funding limitations.
- Comment 10: In page 6 (results): In table 1, there's a note '1' near the number of participants . What does this refer to?
Response 10: Thank you very much for pointing this out. We carefully reviewed and reanalyzed Table 1. Also, we have added footnote and a legend which have been appropriately explained.
- Comment 11: In par 3.1, is it possible to know, or to estimate, which vaccine was/had been in use up to that period in that area.
Response 11: Thank you for the comment. Unfortunately, we cannot evidently show which specific vaccines were administered to participants. However, at the time of the study, the Oxford-AstraZeneca vaccine was largely available to the Ghanaian population which, was primarily distributed through the COVAX initiative.
- Comment 12: Still in par 3.1, it would be interesting to add, if available, the information about 'previously reported COVID-19'.
Response 12: Thank you for your comment. Unfortunately, our data does not specifically capture previously reported COVID-19 cases. We only recorded responses to questions such as ‘Have you had contact with a suspected COVID-19 case?’ and ‘Have you had contact with anyone with flu-like symptoms in the last seven days?’ However, the majority of responses were either unknown or negative, limiting meaningful associations. We acknowledge the importance of this information and will consider incorporating it into future research.
- Comment 13: still in par 3.1, the proportion of people declaring no adherence to restrictions is also interesting and could also be highlighted (as much as the total no+low adherence, which forms a wide majority of the population).
Response 13: Thank you very much for this comment. We acknowledge the importance of highlighting the proportion of individuals who reported no adherence to restrictions. In response, we have now explicitly emphasized this in the results, along with the combined proportion of individuals with low and no adherence, which together represent the majority of the study population.
- Comment 14: In par 3.2, please introduce reference to tab2 after seroprevalence and RT-PCR status.
Response 14: Thank you for your comment. We have introduced a reference to Table 2 immediately after discussing seroprevalence and RT-PCR status to improve clarity and alignment with our reported data.
- Comment 15: still in par 3.2, data about association between infection and vaccination status are presented but are not present in any table. Please add
Response 15: Thank you very much for your comment. We have included data on the above comment. Kindy refer to our re-analyzed Table 2.
- Comment 16: In page 7 (results): in par 3.2, line 2, the proportion of omicron variant is stated as 42.86%, but actually the majority of the typed positivities are listed as 'other' variants. Now, this poses a serious doubt over the test interpretation, since it is possible that some of these' others' could be sub-variants of omicron, or minor types still circulating, but it looks unlikely that they could represent the majority of the positive cases. It looks like more probable that the actual omicron and delta proportion is therefore underestimated.
Response 16: Thank you for raising this important point. We acknowledge that the high proportion of cases categorized as 'other' variants is a limitation of our study. It is possible that this 'other' variants includes Omicron sub-variants or other minor circulating lineages that were not specifically identified due to sequencing constraints. As a result, the reported proportion of the Omicron variant (42.86%) may indeed be an underestimation, as you suggest. Ideally, further sequencing would be performed to fully characterize these 'other' variants and provide a more accurate representation of variant proportions. We have added this as a limitation to the study.
- Comment 17: Maybe, testing with a different assay would change the proportion dramatically, but probably this is no more possible. So, my suggestion is to go more in detail into understanding the technical issue beyond the assay's odd performance, and understanding if, maybe, only the omicron+delta cases correctly genotype could be used for the calculation of the relative proportions, considering the others as 'not genotyped' rather than 'other genotypes'.
Response 17: Thank you very much for your comment. We acknowledge the potential impact of the assay performance. While reanalyzing with a different assay is no longer feasible, we have carefully considered your suggestion.
Comment 18: in page 9 (results): In par 3.3, as shown also in detail in table 3, the role of vaccination status towards seropositivity rate is obviously relevant. In all other variables taken into account, there could be a bias due to that, since it is possible that vaccination status is unbalanced in specific age classes, type of settlement, and more in general in all social classification groups. So, it would be more informative to add a stratification of the other variables with respect of vaccination status, or to take into consideration, for relevant variable, also to show results from 'vaccine-free' population.
Response 18: Thank you very much for this comment. We recognize the value of this approach and will consider it in future research analyses. .
- Comment 19: I understand that seroprevalence is a cumulative concept, but the association with other factors has very different interpretation whether the seroprevalence is due to natural infection or vaccines.
Response 19: Thank you for raising this important point. Our analysis examined the overall association of variables with seroprevalence without distinguishing between antibody responses from natural infection and those induced by vaccination. We acknowledge that this distinction could provide more nuanced insights. Unfortunately, our dataset did not allow for a clear differentiation between these sources of seropositivity.
- Comment 20: In page 9 (discussion): the high seroprevalence is said to be confirmed in other African countries, but ref 17,30 provides very different results.
Response 20: Thank you very much reviewer. We acknowledge that seroprevalence rates can vary significantly across different regions and populations within Africa due to factors such as population density, timing of data collection, local outbreaks, and access to testing. While (references 17 and 30 now reference 29) provide valuable insights into specific locations in Ghana and Togo, respectively, our statement aimed to reflect a broader trend of high SARS-CoV-2 seroprevalence observed in various studies across the continent, including those conducted after the period covered by the cited references.
- Comment 21: The strong association between vaccination and seroprevalence quite obvious, and it is clear that seroprevalence studies performed during the vaccination era are completely different from the first ones, based entirely on natural immunity.
Response 21: Thank you very much for your comment. We cannot agree less. We fully acknowledge that seroprevalence studies conducted during the vaccination era present a fundamentally different context compared to those conducted before vaccine rollouts, which relied solely on natural immunity. The strong association observed between vaccination and seroprevalence in our study highlights this shift.
- Comment 22: The strong correlation with age in ref 31 is found on vaccinated subjects. On non vaccinated it is shown to be the opposite. In this study, the large majority of seropositives are non-vaccinated, so the trend is very different than in ref 31.
Response 22: Thank you very much for your comment. Reference 31 now reference 30 (Busch et al., 2022) was cited to juxtapose the trend of increasing seropositivity among older individuals. However, we acknowledge that their findings were based on a vaccinated population, whereas in our study, the majority of seropositive participants were unvaccinated.
- Comment 23: In page 10 (discussion): - The first period (starting actually from the end of page 9), describes a significant association with pre-existing conditions, but this is excluded in the results. Please check. By the way, the total number of subjects with pre-existing conditions would anyway be too low to get relevant associations.
Response 23: Thank you very much for your comment. You are right that pre-existing conditions were not explicitly included in the results section. Additionally, we acknowledge that the total number of participants with pre-existing conditions in our study is too low to yield statistically meaningful associations.
- Comment 24: - The proportion of acute infections is really high, probably much higher than that reported in the same time period in Americas and Europe. The cited papers (ref 32-34) don't show any similar, nor higher, data. Please check. Also, the GHS bulletins from 2021 seem to show incidence rates below 1%.
Response 24: Thank you very much for your comment. References now reference 31-33.
- Comment 25: In general terms, a 10% positive cases in a random population screening are something quite striking. Reference 36 is completely off-topic.
Response 25: Thank you very much for pointing this out. We acknowledge Reference 36 is not to be relevant in this context. We have removed it accordingly to maintain the focus of the discussion.
- Comment 26: In the references section: - please update the link for reference 1.
Response 26: Thank you very much. We have updated the link for reference 1 accordingly.
- Comment 27: Please check for correct link for reference 2.
Response 27: Thank you very much. We have updated the link for reference 2 accordingly.
- Comment 28: Despite being relevant publications, both ref 3 and 4 are very old; a much more recent selection should be associated to the specific sentence in the text.
Response 28: Thank you very much for your comment.
- Comment 29: Same as above for refs 5 and 6. Coming from the very first year of infection, they're too old to fit the sentence (which deals with 2-years period).
Response 29: Thank you for pointing this out. We have duly updated the refences 5 and 6 to ensure they align with more recent data.
- Comment 30: Ref 10 is actually the announcement of an upcoming paper which is under review in 2022. Please add a current paper link of select a different source.
Response 30: Thank you for pointing this out. We have replaced reference 10 with the relevant, published source to ensure accuracy.
- Comment 31: Ref 18 lacks journal's details. Also, author's name is miswritten
Response 31: Thank you very much for pointing this out. We have updated all noted details for reference 18.
- Comment 32: Ref 20 has a wrong link. Moreover, please describe GHS acronym.
Response 32: Thank you for pointing this out, dearest reviewer. Reference 20 have been duly updated.
- Comment 33: For ref 21, please provide a more specific link.
Response 33: Thank you very much for your comment. Duly rectified. Reference previously listed as reference 21 is now updated to reference 18.
- Comment 34: For ref 24, please provide link or further details for the paper.
Response 34: Thank you very much for your comment. Duly rectified. Reference 24 is now reference 23.
- Comment 35: For ref 25, link for publication is not working or mispelled. Please correct or update.
Response 35: Thank you very much for your comment. Duly rectified. Reference 25 is now reference 24.
- Comment 36: Reference 28 is from a preprint with no peer-review from 2020. please update or select another source.
Response 36: Thank you for pointing this out. Comment has been duly resolved.
- Comment 37: Ref 34 is from a continuously updated page. Please provide a specific link for the bulletin chosen for comparison in the relevant period.
Response 37: Thank you very much for your comment. Duly rectified. Reference 34 is now reference 33.
- Comment 38: Ref 37 title's is mispelled and date is wrong.
Response 38: Thank you for pointing this out. Reference duly updated. Reference 37 is now reference 35.

Round 2
Reviewer 1 Report
Comments and Suggestions for Authors
The manuscript was significantly revised but still contains inaccuracies. The method ELISA is not described properly. Origin, isolation and purification of the SARS-CoV-2 antigen(s) for detection of antibodies remain unclear. Chromogen is not mentioned. Units of measurements are missing.
The revised manuscript lacks the section "Conclusion". The data are not enough to discuss SARS-CoV-2 "population dynamics" (the only time point - December 2021 and the majority of isolates (48.47%) have not been identified (the revised Table 2)).
Problem points are highlighted in the attached file.

English language is not always precise in the revised manuscript.
Reviewer 2 Report
Comments and Suggestions for Authors
I would like to thank the authors for taking into consideration many of my minor points comments to improve their paper. On the other hand, I must notice that no reply has been provided about the major points explained in the first part of my review.
So, I would like to remind the authors that my major concern was about the unclear relationship between the data shown on this paper and those already published as a part of a wider cohort in the previous paper hereby cited as reference 25. In a reply to a minor points the authors state that 'the data presented in this study were not included in the national-level study'. But this seems inconsistent with what is in the main text, i.e. that ' A total of 1027 participants were recruited, consented and sampled. These formed part of a nationwide seroprevalence assessment [25]'
So, if the subjects of the current study 'formed a part of the nationwide assessment', and data from that assessment were already published years ago, at least a part of this study's data were already published too. So I newly invite the authors to make clear and sound statements in the text, both in the methods, results, and discussion, about which part of the methods, results, and associations described here are actually mutauted, or adapted, from the old paper.
Another 'major' suggestion which fell completely ignored, was about trying to recalculate data about the association between serologic status and other variants by adding a second alternative option, i.e., limiting the analysis to non-vaccinated subjects, as to rule out the effects of vaccination, which, in my opinion, is a confonding factor here, since probably vaccination coverage was uneven between subjects belonging to the various callification groups (age, social conditions, geography, restriction adherence, and so on). I renew the suggestion, and of course the new analysis can be added to the new one, it needs not to replace it.
This said, there is still some improvement related to the old minor points needing to be assessed:
- After the merging of some paragraphs, the other paragraphs of chapter 2 must be re-numbered
- reference numbers must be chcked for eventual mistakes due to recent changes. For example, in line 113, the old [28] has been left instead of new [27]. Which, still in this new version, has no link to any specific protocol file or page
- the explanation provided about the ROX aspecific signal in the genotyping assay is sincerely disapponting, as the authors seem to have considered as true positive, yet with unclear genotype status, what now seems to me to be basicly 'bona fide' aspecific background. If it is like that I seriously have doubts about the entire set of results. But, willing to be optimistic, at least, I invite the authors to erase from the text and tables every consideration about this sub-group of cases, limiting their associations and comments to the genotyped (either omicorn or delta) ones, and leveing the rest as "not genotyped".
- In the discussion section, some of the study's findings have been put in comparison with literature papers, but in some cases, as notified in the prior revision, in improper ways, as those papers were saying different or even opposite things (e.g., references 17, 29, 30). I still invite the authors to modify their text as to better define the relationship between their results ando those found by the other papers, including when they are not in accordance
- Also, in their reply, the authors agree with my comment about the description in the discussion of a 'significant portion of subjects with pre-existing conditions'. Yet, they didn't modify the text, so I newly invite them to correct that sentence which is clearly incorrect
- Finally, about the discussion, the authors thank me for my comment about the comparison with data in Europe and America, but once again they left their text unmodified. My comments are not provided for the aim of receiving thanks, but to address the authors in modifying parts of the paper which are identified as 'incorrect', and must therefore be changed accordingly.
- The reference section still has some minor corrections required: Ref 18 has author's name initials before the last name; ref 22 and 24 are a replicate (so please, re-check all ref numbers in the text after the correction), furthermore, the provided link is incorrect or dead; ref 23 has a mistake in the author's name and provides no details about the journal, nor links or doi; ref 27 provides no link, nor authors info, and the associated doi is wrong (it belongs to the previous version's ref 28); ref 32 has the author's name mispelled (S.B. are middle names, not part of the last name); as already stated in the previous revision, ref 33 needs to be associated to some time-set info, or to a timed link, since it is a continuosly updated page, and if checked now it 'll provide info different from those reported in the paper
